# Cavity-enhanced single-shot readout of a quantum dot spin within 3 nanoseconds

Nadia O. Antoniadis [1,4], Mark R. Hogg [1,4] ✉, Willy F. Stehl[1], Alisa Javadi [1,3], Natasha Tomm [1], Rüdiger Schott [2], Sascha R. Valentin [2], Andreas D. Wieck [2], Arne Ludwig [2] & Richard J. Warburton [1] ✉

Rapid, high-fidelity single-shot readout of quantum states is a ubiquitous requirement in quantum information technologies. For emitters with a spin-preserving optical transition, spin readout can be achieved by driving the transition with a laser and detecting the emitted photons. The speed and fidelity of this approach is typically limited by low photon collection rates and measurement back-action. Here we use an open microcavity to enhance the optical readout signal from a semiconductor quantum dot spin state, largely overcoming these limitations. We achieve single-shot readout of an electron spin in only 3 nanoseconds with a fidelity of (95.2 ± 0.7)%, and observe quantum jumps using repeated single-shot measurements. Owing to the speed of our readout, errors resulting from measurement-induced back-action have minimal impact. Our work reduces the spin readout-time well below both the achievable spin relaxation and dephasing times in semiconductor quantum dots, opening up new possibilities for their use in quantum technologies.

The ability to perform a projective measurement of a quantum state in a single measurement (single-shot readout) is an enabling technique in quantum technologies[1,2]. Single-shot readout is necessary in quantum computation in order to extract information at the end of the protocol, as well as in error detection and correction as the quantum processor runs[3]. In addition, single-shot readout is necessary to close the fair-sampling loophole in tests of quantum non-locality, and was a key ingredient in recent demonstrations of loophole-free Bell inequality violations[4]. The ideal single-shot readout protocol achieves high-fidelity qubit readout in the shortest time possible; readout within the qubit dephasing time is essential for quantum error correction, and enables measurement-based quantum feedback[5,6].

The spin states of semiconductor quantum dots (QDs) show exceptional promise in quantum technology[7–9]. Optically active QDs, established bright and fast sources of coherent single photons[10–13], can be occupied with a single electron and the electron spin can be initialised[14,15] and rotated on the Bloch sphere[16,17] on nanosecond time-scales using all-optical techniques. Theoretical proposals[18,19] and recent

experiments[20–22] have established the spin-photon interface provided by the InGaAs platform as a leading contender for creating photonic cluster states, an important resource for quantum repeaters[23] and measurement-based quantum computation[24]. The dephasing time of the electron spin in optically active QDs is limited by magnetic noise arising from the nuclear spins. However, there are powerful mitigating strategies. A double-QD can be used to create a clock-transition[25]; a switch to a hole spin suppresses the effect of the magnetic noise particularly in an in-plane magnetic field[26,27]; and the noise can be almost eliminated by laser-cooling the nuclei[28,29]. In the context of cluster states, spin readout is necessary in order to disentangle the spin from the photons, thereby releasing an entirely photonic entangled state. To date, single-shot spin readout on a timescale comparable to the rapid spin initialisation and manipulation times has remained elusive.

Spin readout with an optical technique typically proceeds by applying a magnetic field to a QD containing a single electron, reso-nantly driving one of the Zeeman-split trion transitions, then collecting the spin-dependent resonance fluorescence[30]. However, during readout,

[1]Department of Physics, University of Basel, Klingelbergstrasse 82, CH-4056 Basel, Switzerland. [2]Lehrstuhl für Angewandte Festkörperphysik, Ruhr-Universität Bochum, D-44780 Bochum, Germany. [3]Present address: School of Electrical and Computer Engineering, Department of Physics and Astronomy, The University of Oklahoma, 110 West Boyd Street, Norman, OK 73019, USA. [4]These authors contributed equally: Nadia O. Antoniadis, Mark R. Hogg. ✉e-mail: mark.hogg@unibas.ch; richard.warburton@unibas.ch

the applied laser can induce an unwanted spin flip[31,32], a process known as back-action. The key challenge for spin readout is to collect enough photons to determine reliably the spin state before the back-action flips the spin. Of the small number of previous experiments to achieve single-shot readout of InGaAs QD spin states[33–35], the most rapid to date achieved a fidelity of 82% in a readout time of 800 ns[34]. This 800 ns readout time was similar to the back-action timescale, and is significantly longer than the dephasing time for an electron spin bound to an InGaAs QD ($T_2^* = 125$ ns following nuclear bath cooling[28,29]).

In this work, we report nanosecond-timescale, all-optical, single-shot spin readout. We use an open microcavity to boost the photon collection efficiency in order to reduce the spin readout time. We achieve single-shot readout in only 3 nanoseconds with a fidelity of $(95.5 \pm 0.7)\%$, an improvement in readout speed of more than two orders of magnitude with respect to previous experiments. To the best of our knowledge, this is the fastest single-shot readout of a quantum state ever achieved across any material platform. Our approach brings the readout time well below the dephasing time for an electron spin in this system. Cavity enhancement is a powerful tool for improving optical single-shot spin readout in other systems; photonic crystal cavities have been successfully used with defect centres in diamond[36] and rare-earth ions[37,38]. Importantly our open microcavity approach is not specific to QD samples, and can be used to enhance optical spin readout in other material platforms[39].

## Results

### High-efficiency photon collection

A schematic of the setup used in our experiments is shown in Fig. 1a. Our sample is a gated, charge-tunable InGaAs/GaAs device, with a highly reflective Bragg mirror integrated into the semiconductor heterostructure[13,40]. The gate structure allows the charge occupancy of the QD to be set, as well as fine tuning of the emission frequency via the

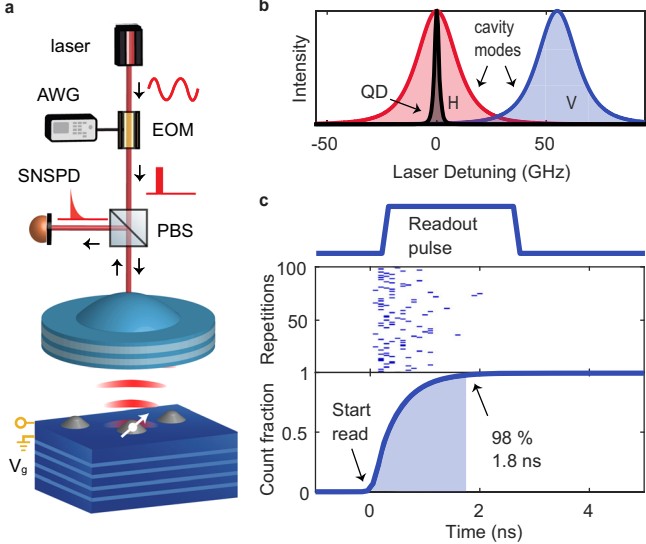

**Fig. 1 | Experimental setup and system efficiency. a** Resonant laser pulses with variable intensity and duration are sent to the QD using an electro-optic modulator (EOM) driven by a fast arbitrary waveform generator (AWG). The photons emitted by the QD are collected in the output arm of the cross-polarised microscope and measured on an SNSPD (superconducting nanowire single photon detector).
**b** Frequency configuration of the QD and mode-split cavity with respect to the laser. **c** Readout characterisation at zero magnetic field: here, the readout pulses are set to a duration of 2 ns (top panel) with a repetition time of 100 ns. Photons emitted by the QD are detected and the arrival times registered for 100,000 repetitions of the pulse sequence; 100 example traces are depicted in the middle panel where the blue dots represent a photon detection event. In 98% of the repetitions, a photon is detected within 1.8 ns.

quantum-confined Stark effect. We operate with a single electron occupying the QD, which is our spin readout target. A miniaturised Fabry–Pérot cavity is created between the semiconductor bottom mirror and a free-standing concave top mirror. The QD sample is attached to an XYZ nano-positioning stage. This flexibility of the open microcavity design allows the cavity to be re-positioned to address a chosen QD. Once a QD is positioned at the anti-node of the cavity field (XY positioning), its frequency can be matched to one of the QD transitions (Z positioning).

Figure 1c demonstrates the high photon collection efficiency of our microcavity system and its potential for rapid spin readout. A photon emitted by the QD exits the output facet of the collection single-mode fibre with a 57% probability[13]. The overall system efficiency, $\eta$, the probability that an exciton in the QD results in a click on the detector, is 37%. Initially, we set the magnetic field to zero, such that the optical transitions for both electron spin states are degenerate. In this scenario, a resonant laser pulse excites the QD optical transition regardless of the electron spin state. The readout pulse drives the optical transition, and the QD emits photons at a rate set by the (Purcell-enhanced) optical decay rate. The time required for a photon emitted from the QD to be registered by the detector depends on the overall system efficiency; for high efficiencies, a photon is rapidly detected. We apply a train of 2 ns readout pulses (approximately square temporal shape limited by the 350 ps AWG rise time, separated by 100 ns, with an optical power equal to six times the QD saturation power) to the QD, and monitor the collected photons on the single photon detector (an SNSPD). The SNSPD has a dead time of ~12 ns, meaning that after one photon has been detected another detection event stemming from the same pulse is extremely unlikely. Thus, although the QD emits at a constant rate during the 2 ns readout pulse, a maximum of one photon detection event occurs. We repeat the pulse sequence 100,000 times, and analyse the fraction of pulses in which a photon was detected as a function of the readout duration. Our detector registers the precise arrival time of each photon detected during the 2 ns readout pulse, which we use to plot the probability of detecting a photon as a function of elapsed readout pulse duration (see bottom panel Fig. 1c). We find that for 98% of the traces, a photon is detected within 1.8 ns. When the same pulse sequence is repeated with the QD detuned out of resonance with the readout laser, we detect a photon (due to laser leakage within the cross-polarised setup) for <0.1% of the pulses, demonstrating that the photons we detect are almost exclusively created by the QD.

### Single-shot spin readout

To perform single-shot spin readout, we apply a magnetic field of 2.0 T along the growth direction of the sample (Faraday configuration), which creates a four-level system in which the two strongly allowed trion transitions with linewidth $\Gamma/(2\pi) = 2.8$ GHz (which corresponds to the transform limit) are split by 55 GHz (the sum of the electron and hole Zeeman splittings, 6.8 and 20.7 GHz/T respectively). Spin readout is achieved by tuning the cavity into resonance with one of the strongly allowed transitions, as shown in Fig. 2b. The readout pulse sequence is then similar to that shown in Fig. 1c, but photon emission is now only enhanced for the trion transition resonant with the cavity. Figure 2a shows an example of single-shot readout traces: here, we apply a train of readout pulses (5 ns duration with a repetition time of 100 ns) resonant with the cavity-enhanced $|\uparrow\rangle \leftrightarrow |\Uparrow\Downarrow, \uparrow\downarrow\rangle$ trion transition. We note that for this experiment, the frequency alignment of cavity modes and trion transitions is identical to that shown in Fig. 2b. If the electron is projected into the $|\uparrow\rangle$ spin state, Purcell-enhanced fluorescence from the $|\uparrow\rangle \leftrightarrow |\Uparrow\Downarrow, \uparrow\downarrow\rangle$ trion will be rapidly registered by the detector. The spin is thus projected into the bright state, and detecting a single photon emitted by the QD during the readout pulse constitutes a measurement of the spin state. Conversely, if the electron is projected into the $|\downarrow\rangle$ (dark) spin state no fluorescence is detected, as the $|\downarrow\rangle \leftrightarrow |\Downarrow\Uparrow, \uparrow\downarrow\rangle$ trion is out of resonance with the readout laser. In this case, the

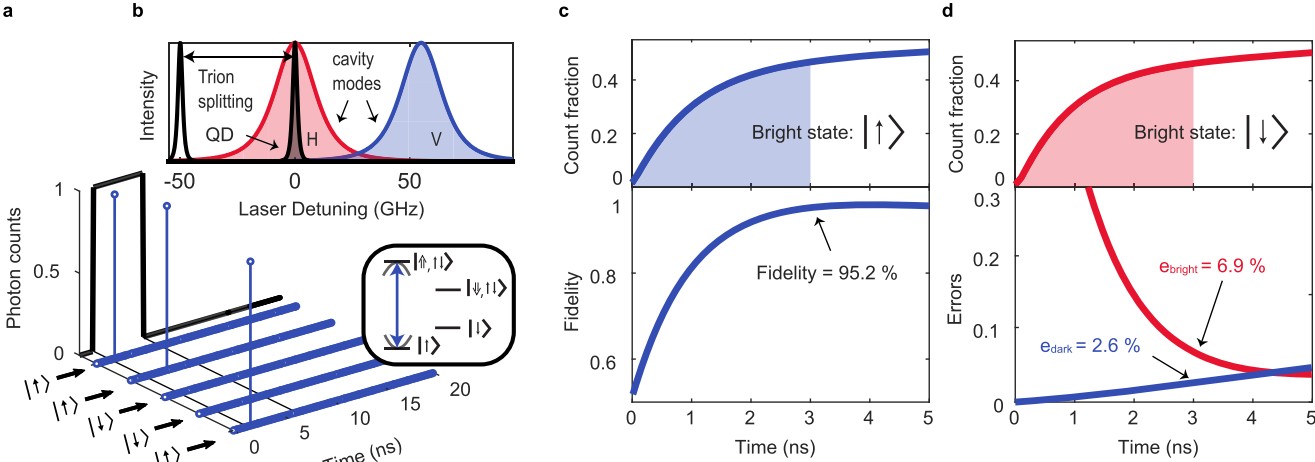

**Fig. 2 | Single-shot readout of the QD spin at 2.0 T. a** Example single-shot readout traces. If a photon is detected during the readout pulse, the state of the QD is assigned to the bright state (here, spin up $|\uparrow\rangle$). Repetitions with no detected photon are assigned to the dark state (here, spin down $|\downarrow\rangle$). Schematic of the QD energy levels in a magnetic field, indicating the readout transition (here, bright state $|\uparrow\rangle$, blue arrow) and the cavity frequency. **b** Frequency configuration of the two QD transitions and mode-split cavity with respect to the laser. With a 2.0 T magnetic field, only one trion transition is resonant with the H-polarised cavity mode, resulting in spin-selective Purcell enhancement. **c, d** Experimental count fraction (top) and corresponding readout fidelity/errors (bottom) as a function of readout time for the bright state being up/down. Here, readout pulses with a duration of 5 ns are used. The pulse sequence is repeated 100,000 times. We achieve a readout fidelity of 95.2% for a readout time of 3 ns.

absence of a detector event during the readout pulse indicates that the spin was projected into the dark state. We stress again that the readout time is less than the dead time of the detector: a maximum of one photon can be measured during the readout process. Furthermore, the overall system efficiency is high enough that the absence of a detected photon contains significant information: it denotes that the spin was projected into the dark state. The detection is thus binary: detection of one photon corresponds to the $|\uparrow\rangle$ state, and zero photons to the $|\downarrow\rangle$ state. Equivalently, our photon number threshold for discriminating the spin states is one single photon.

We repeated the spin-readout measurements with the cavity and readout laser tuned such that either $|\uparrow\rangle$ or $|\downarrow\rangle$ is the bright transition. In Fig. 2c we show the results of 100,000 repetitions of the spin readout pulse sequence with $|\uparrow\rangle$ set as the bright state (the configuration shown in Fig. 2b). We plot the fraction of readout traces containing one photon, i.e. the fraction of traces we assign the electron spin state to be $|\uparrow\rangle$. We observe a rapid increase in the count fraction (on a timescale of a few nanoseconds) as a function of the readout time. Compared to the 0 T results in Fig. 1c, the maxima of the count fractions now saturate close to 50%: each spin state is almost equally likely. The reason is that the spin is not initialised in these experiments. Instead, before readout, the spin is in a mixed state as co-tunnelling between the QD and the Fermi sea of the back contact regularly randomises the spin state (on a timescale of ~150 ns) during the 100,000 readout pulse repetitions, such that both $|\uparrow\rangle$ and $|\downarrow\rangle$ spin states have approximately equal probabilities. We note that spin initialisation via optical pumping was possible in our experiment, albeit with a modest fidelity of approximately 67% due to the relatively rapid co-tunnelling rate (Supplementary Note 2). Figure 2d shows data for 100,000 repetitions of the readout pulse sequence, now with the readout laser resonant with the low-frequency trion transition, $|\downarrow\rangle \leftrightarrow |\downarrow\uparrow\downarrow\rangle$ (thus making the $|\downarrow\rangle$ state the bright state and $|\uparrow\rangle$ the dark state).

Compared to the 0 T readout in Fig. 1c, the readout speed is slightly slower (high-fidelity readout is achieved in 3 ns rather than 1.8 ns). The reason for this slower readout is that at 2.0 T we operate with the laser on resonance with the QD but detuned by 7.5 GHz from the actual cavity resonance, where we observe optimal laser suppression at the cost of a reduced Purcell factor ($F_P = 6.1$ compared to $F_P = 8.5$ exactly at resonance). Consequently, the readout speed is slightly

reduced compared to 0 T. However, we still achieve high-fidelity single-shot spin readout within 3 ns.

In order to estimate the spin-readout fidelity, we perform Monte-Carlo simulations of the single-shot traces with parameters matching our experiment. The simulations include only a few parameters: the overall system efficiency $\eta$, the Purcell factor $F_P$, and the spin-flip time, i.e. the relaxation time, $T_1$. At $B = 0$, $\eta = 37\%$. At $B = 2.0$ T, technical issues result in a slightly reduced efficiency, $\eta = 25\%$ (Supplementary Note 4). The spin $T_1 = 158$ ns was measured via the quantum jump experiments discussed in the next section. We define the readout-time-dependent fidelity as[33]

$$\mathcal{F}(t) = 1 - p_{\text{bright}} \cdot e_{\text{bright}}(t) - p_{\text{dark}} \cdot e_{\text{dark}}(t), \quad (1)$$

where $p_{\text{bright}}$ ($p_{\text{dark}}$) is the occupation probability of the bright (dark) state, and $e_{\text{bright}}$ ($e_{\text{dark}}$) the respective time-dependent probability of assigning the spin state incorrectly. The spin occupation probability distribution depends on the spin-flip rates, as well as the readout pulse duration and repetition rate; for our experiments, it is approximately 50:50 ($|\uparrow\rangle : |\downarrow\rangle$). The error $e_{\text{bright}}$ is determined on these timescales by imperfect overall system efficiency (which can lead to a spin projected into the bright state being incorrectly assigned as the dark state should no photon be detected). The error $e_{\text{dark}}$ is determined by laser leakage (which can lead to a spin projected into the dark state being incorrectly assigned as the bright state). Errors due to spin-flips during the readout time (either due to laser back-action or spin relaxation) play a minor role in our experiment. Our Monte-Carlo simulations capture all of these error sources quantitatively ($e_{\text{dark}} = 2.6\%$, $e_{\text{bright}} = 6.9\%$ at 3 ns; full details of the fidelity calculation and the influence of readout errors can be found in Supplementary Note 5). The simulated count fractions show very good agreement with our experimental results and allow us to extract a maximum readout fidelity of $(95.2 \pm 0.7)\%$ in 3 ns. The calculated readout fidelity and associated readout errors as a function of readout-time for the configuration with $|\uparrow\rangle$ as the bright state is plotted in the lower panel of Fig. 2c, d, respectively.

## Repeated readout and quantum jumps

The fast spin readout enables us to probe the electron spin dynamics. By repeated single-shot measurements of the spin state, we can

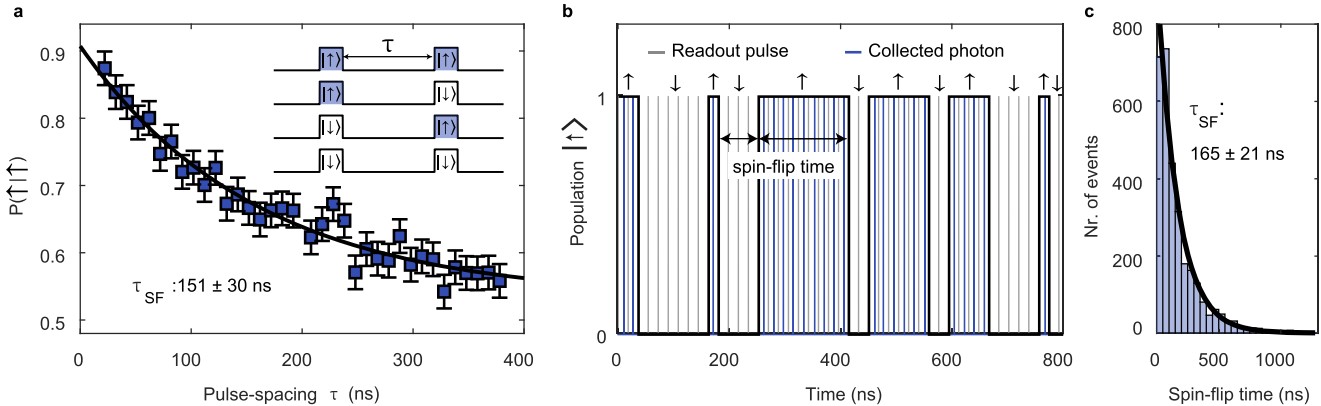

**Fig. 3 | Repeated single-shot measurements and quantum jumps. a** Conditional probability of measuring $|\uparrow\rangle$ given that the first measurement returned $|\uparrow\rangle$ for two sequential (3 ns duration) readout pulses as a function of $\tau$, the delay between the pulses (the errorbars are one standard deviation). For short values of $\tau$ the second measurement outcome is correlated with the first; for longer values of $\tau$, the probability to acquire the same outcome decreases exponentially, revealing a spin-flip rate of $151 \pm 30$ ns. **b** By repeatedly measuring the spin state with excitation pulses spaced by 18 ns (above the detector's dead time), we observe quantum jumps of the spin state. **c** The time between the spin flip events in (**b**) are extracted for a total measurement time of 2.4 ms and are summarised in a histogram. The distribution of the events reveals a spin-flip time of 165 ns, matching well the result of the two-pulse measurements in (**a**).

determine the spin-flip time from the correlation between sequential measurements. In addition, we can track the electron spin state in real time, observing quantum jumps as the spin flips. In Fig. 3a we perform a pulse sequence consisting of two readout pulses separated by a time $\tau$. Here we fix the length of both readout pulses to be 3 ns, and the pulse repetition time to be 400 ns. The first readout pulse is a projective measurement of the spin state: in effect, the spin is initialised at $\tau = 0$ with a fidelity given by either $e_{bright}$ or $e_{dark}$. This approach of initialisation-by-readout provides an alternative to spin initialisation via optical pumping, and in our experiments results in a higher spin initialisation fidelity ($F_{init}^{|\uparrow\rangle} = 93.1\%$ and $F_{init}^{|\downarrow\rangle} = 97.4\%$ with $|\uparrow\rangle$ as the bright state, compared to $F_{init}^{|\uparrow\rangle,|\downarrow\rangle} \sim 67\%$ for optical pumping). The second readout pulse can then be used to determine the spin state at $\tau > 0$ allowing us to measure the correlation between the two measurement outcomes as a function of $\tau$. Figure 3a shows the conditional probability of measuring spin $|\uparrow\rangle$ in the second pulse (as a function of $\tau$), given that the first read result returned $|\uparrow\rangle$. We note that the minimum spacing between the two pulses is limited to $\tau \gtrsim 12$ ns by the dead time of the detector. Increasing $\tau$ decreases the probability of reading out the same spin state for both pulses due to spin flips, and for large $\tau$ the second read is completely uncorrelated with the first. By fitting an exponential decay to the data in Fig. 3a, we extract a spin-flip time of $150 \pm 30$ ns. Furthermore, the limit as $\tau \to 0$ of this conditional probability is approximately $1 - e_{bright}$, confirming the value of $e_{bright}$ determined from the Monte-Carlo simulations. Similarly, a measurement of the dark-dark conditional probability confirms the value of $e_{dark}$.

Given that our readout sequence is much shorter than the spin lifetime, we can use repeated single-shot measurements to detect real-time quantum jumps of the electron spin state. For that purpose, we send in a train of 3 ns readout pulses spaced by 18 ns, slightly above the minimum allowed by the detector's dead time. We observe quantum jumps in the spin state, as shown in Fig. 3b. (In the original quantum jump experiment, the quantum jumps between the bright and dark states were driven with weak coherent excitation[41]. Here, the jumps are driven by a dissipative process, energy exchange with the Fermi sea via co-tunnelling.) The time between spin-flip events during a 2.4 ms total acquisition period is extracted and summarised in the histogram in Fig. 3c. From the exponential decay in the number of events per flip time, we can extract the spin-flip time to be approximately 165 ns, consistent with the results from the double-pulse experiment in Fig. 3a.

## Discussion

We have demonstrated that the frequency-selective Purcell enhancement provided by our optical microcavity enables us to perform single-shot readout of a QD spin state within a few nanoseconds, with fidelity as high as 95%. Our results bring the spin readout time for semiconductor QDs close to the short optical spin manipulation times[16,17], and well below previously demonstrated relaxation ($T_1$)[42] and dephasing ($T_2^*$) times[17,28,29]. For recent loophole-free Bell tests, entangled nitrogen vacancy (NV) centres were positioned 1.28 km apart to allow 4.27 μs for the Bell sequence to be performed such that the NVs are space-like separated[4]. Of this 4.27 μs, 3.7 μs (corresponding to 1.1 km) was used for the single-shot spin readout. Our rapid spin readout indicates that similar Bell tests could be performed using semiconductor QDs located significantly closer together, mitigating the challenge of synchronising experiments between different buildings; the separation distance enforced by our readout time is less than one metre. By combining the highly indistinguishable photons created by remote semiconductor QDs[43], the high system efficiency of our microcavity[13], along with $T_2^*$-enhancement via cooling of the nuclear spins[28,29], high-fidelity spin-spin entanglement generation rates of a few hundred MHz are feasible.

We can foresee several ways to improve the readout time in our experiment even further. Most simply, the overall system efficiency can be increased by improving the detector system (fibre couplers and detector itself). Furthermore, it should be possible to operate at the true cavity resonance in an applied magnetic field, thereby at maximum Purcell factor. Our Monte-Carlo simulations show that these changes would allow single-shot readout with a fidelity of 99.5% in less than one nanosecond to be achieved.

We also note that although spin initialisation via optical pumping was not required for our present experiments, the ability to initialise the spin in a specific target state will likely be important for future quantum technological applications. Spin initialisation via optical pumping has been achieved with fidelities $\geq 99.8\%$ in similar QD samples[14], and should be achievable in our system by increasing the tunnel barrier thickness between the QD layer and the n-doped Fermi sea to reduce the co-tunnelling rate, which limited our initialisation fidelity to approximately 67% in the present experiments. Future experiments using devices with a larger tunnel barrier are strictly necessary to verify that high-fidelity spin initialisation can be combined with cavity-enhanced single-shot readout. Based on the detailed understanding of cotunneling[42], chances of success are very high.

Given the efficient generation of single photons, fast spin initialisation and rotation, and now fast single-shot spin readout, the next step is the implementation of coherent manipulation of the spin-state together with spin readout. Fast spin manipulation relies on a Raman transition that is naturally established in an in-plane magnetic field (Voigt configuration). In this case, the four transitions (Fig. 2b) have equal optical dipole moments such that readout back-action is maximal: spin readout becomes challenging. With our approach, this longstanding problem, spin readout in the Voigt geometry, can be solved: the resonant cavity restores a spin-conserving process, i.e. a cycling transition; the high overall system efficiency enables a readout outcome before back-action occurs. With our present Purcell enhancement, our simulations show that in the Voigt configuration, readout on the same timescale with a fidelity of up to 89.9% is feasible.

Coherent spin control of InGaAs QDs typically uses a Raman laser detuned by $\geq 200$ GHz from the QD resonance; in our cavity system, this laser can access the QDs via a waveguide-like mode propagating perpendicular to the cavity axis along the sample surface. This side-excitation strategy has previously been used to demonstrate resonance fluorescence in similar QD samples[44], and is conceptually similar to the side-access control used in trapped atom-cavity systems[45]. Alternatively, the Raman transition can be driven directly through the cavity; field enhancement inside the cavity is proportional to $\sqrt{F}$ (where $F = 500$ is the cavity finesse), and for our one-sided cavity the intra-cavity field is enhanced out to detunings of $\Gamma_c\sqrt{F/\pi} = 315$ GHz (where $\Gamma_c = 25$ GHz is the cavity linewidth), suitable for coherent spin control. Our cavity platform is thus capable of integrating rapid QD spin control, readout and photon emission simultaneously, and is a potentially exceptional platform for spin-photon technologies such as the generation of photonic cluster states.

## Methods

### Device and cavity structure
The semiconductor heterostructure containing the QDs is an n-i-p diode structure grown by molecular beam epitaxy. The n-i-p diode is grown on top of a highly reflective Bragg mirror, consisting of 46 $\lambda/4$-layer pairs of AlAs/GaAs, with a nominal centre wavelength of 940 nm (the actual measured centre wavelength is 917 nm). This Bragg reflector is the bottom mirror of the open microcavity. The n-i-p diode consists of a 41.0-nm Si-doped GaAs layer ($2 \times 10^{18}$ cm$^{-3}$ doping concentration), a 25.0 nm undoped GaAs layer that forms the tunnel barrier between the n-doped electron reservoir and the QD layer, a QD layer (self-assembled InGaAs QDs are grown by the Stranski-Krastanov process then capped with 8.0 nm of GaAs), a blocking barrier (190.4 nm of Al$_{.33}$Ga$_{.67}$As) and a p-contact (5.0 nm C-doped GaAs, $2 \times 10^{18}$ cm$^{-3}$ followed by 20.0 nm $1 \times 10^{19}$ cm$^{-3}$). The p-contact is capped with a further 54.6 nm of GaAs, which is passivated with an 8 nm layer of Al$_2$O$_3$.

In our present device, the 25.0-nm tunnel barrier results in spin relaxation dominated by co-tunnelling between a QD electron and the electrons at the Fermi energy in the back contact[42,46] and results in the relatively short $T_1$-times measured here (see Supplementary Note 1 for details). We note that the co-tunnelling rate can be decreased exponentially by increasing the thickness of the tunnel barrier and very long $T_1$-times (approaching a second) have been recorded at the magnetic fields used here[42]. In future experiments a long $T_1$ time is desirable in order to implement nuclear cooling strategies[29]; however, in the present exploratory experiments, the short $T_1$ times were useful in that co-tunnelling randomises the nuclear spins thereby avoiding optical dragging[47].

The top mirror of the tunable microcavity is fabricated using CO$_2$-laser ablation of a fused silica substrate to create a concave mirror shape with a radius of curvature $R = 12$ μm. After laser ablation, the top mirror is coated with 8 $\lambda/4$-layer pairs of Ta$_2$O$_5$ ($n = 2.09$ at $\lambda = 920$ nm)

and SiO$_2$ ($n = 1.48$ at $\lambda = 920$ nm). The number of layer pairs was designed to maximise the photon extraction efficiency from the system[13]. The cavity exhibits a mode splitting between orthogonal horizontal (H) and vertical (V) polarised modes due to a slight birefringence in the semiconductor sample (50 GHz in our experiments). Each cavity mode has a Q-factor of 16,000.

### Optical measurement setup
We use a cross-polarised microscope setup to separate the resonant excitation laser light from the QD emission[48]. The mode splitting between the orthogonally polarised modes of our microcavity allows us to excite the QD using the Lorentzian tail of one mode and to collect via the other mode[13,49]. The QD photons emitted from the cavity are then elliptically polarised, with the major polarisation axis orthogonal to the excitation laser polarisation. Using this technique, we overcome what would otherwise be a 50% loss of QD photons in the cross-polarisation optics[49].

At 0.0 T we measure a Purcell factor of $F_P = 8.5$ (see Supplementary Note 4), where $F_P$ is the enhancement of the emitter's decay rate in the cavity compared to the decay rate in the bare sample. At 2.0 T we observe poor laser suppression in the cross-polarised microscope head at the exact cavity resonance, which we attribute to a Faraday effect in the objective lens and/or cavity top mirror. However, the laser suppression is recovered at a frequency detuned by 7.5 GHz from the exact cavity resonance (see Supplementary Note 4 for further details). For spin readout, we operate at optimal laser suppression where the Purcell factor is reduced to $F_P = 6.1$.

We generate short (1–10 ns) optical readout pulses using a high-bandwidth EOM (EOSpace AZ-6S5-10-PFA-SFAP-950-R5-UL) driven by a high sampling-rate AWG (Tektronix 7122C). Photons emitted from the QD are detected using a superconducting nanowire single photon detector (SingleQuantum, efficiency 82% at 920 nm), and detection events are registered using a time-correlated single photon counting module (Swabian Instruments Timetagger Ultra).

The full cavity stack is mounted in a liquid helium cryostat operating at 4.2 K, equipped with a superconducting solenoid magnet. We operate at 2.0 T for the spin readout experiments.

## Data availability
The data presented in the main text can be downloaded from https://doi.org/10.5281/zenodo.7937818.

## Code availability
The code that supports the findings of this study is available from the corresponding authors upon reasonable request.

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

## Acknowledgements

We acknowledge financial support from Horizon 2020 FET-Open Project QLUSTER, Swiss National Science Foundation project 200020_204069, and NCCR QSIT. A.J. acknowledges support from the European Union's Horizon 2020 Research and Innovation Programme under Marie Skłodowska-Curie grant agreement no. 840453 (HiFig). S.R.V., R.S., A.L. and A.D.W. gratefully acknowledge support from DFH/UFA CDFA05-06, DFG TRR160, DFG project 383065199 and BMBF Q.Link.X.

## Author contributions

N.O.A., M.R.H. and W.F.S. performed the experiments with input from A.J. and R.J.W. N.T. assembled the cavity structure and fabricated the silica top mirror. R.S., S.R.V., A.D.W. and A.L. fabricated and processed the semiconductor device. N.O.A. and M.R.H. performed the analysis with input from A.J. and R.J.W. N.O.A., M.R.H. and R.J.W. wrote the paper with input from all authors.

## Competing interests

The authors declare no competing interests.

## Additional information

Mark R. Hogg or Richard J. Warburton.

