## [Peer Review File · Nature Communications]

REVIEWERS' COMMENTS

Reviewer #1 (Remarks to the Author):

The authors have satisfied my concerns and, in my view, the comments from the other reviewers. The authors report a significant advance that should be published in Nature Comm. without further revision. Notably, they have expanded the discussion of spin initialization in the main text and have corrected one of my points concerning the relevance of T. Nowozin et al. [Appl. Phys. Lett. 104, 053111 (2014)].

Reviewer #2 (Remarks to the Author):

The authors addressed well all the comments. A couple of final points:

1. I agree that fast spin readout is of value to the community, even though it is demonstrated here on a structure with a very strong cotunneling. But I can't agree when the authors say that short electron spin T_1 is totally unrelated to the applicability of the readout method. This statement is a speculation until it is proven experimentally. The proximity of the electron Fermi sea, manifested in short T_1 , has a variety of other possible consequences. One is already appreciated by the authors – the nuclear spin dragging induced by the resonant optical pumping. A thicker barrier, required to achieve a usefully long electron spin qubit T_1 , may also result in less controllable dynamics of the trapped charges. One may speculate that these and other manifestations of the solid-state environments will not affect the cavity-enhanced readout. In my view this would be a plausible speculation, but speculating is not the same as demonstrating.

I strongly suggest that the authors make this distinction explicit and clear, by explaining that the performance of this readout method in structures with weak cotunneling is subject to further experimental verification. This could be done for example in the paragraph around line 354.

2. “Our open microcavity approach can be used to enhance optical spin readout in other systems, such as nitrogen vacancy centres in diamond [36]”. I would argue it has been done already, for example in Science 362, 662 (2018). Moreover, optical cavities have already been used to enhance the readout of the quantum dot spins [Phys Rev Appl 9, 054013 (2018)]. The principle of improving spin readout through cavity-enhanced cyclicity has been applied in other solid state systems, for example in Nature Communications 11, 1605 (2020). I suggest the authors do a bit more work in terms of putting their results into context. This way research done by others would get the due credit, and this paper will have a broader impact.

Otherwise, the paper is well written, and the result is nice. Once the aforementioned issues are rectified, the work can be recommended for publication in Nature Communications.

Reviewer #3 (Remarks to the Author):

The authors address all the comments from the previous review reports. I am glad to recommend its publication.

June 8, 2023

Re: NCOMMS-23-08750-T revision, **Cavity-enhanced single-shot readout of a quantum dot spin within 3 nanoseconds**, Nadia Antoniadis *et al.*

Please find attached our response letter addressing the final Reviewer comments for our manuscript NCOMMS-23-08750-T. We again thank the Reviewers for their efforts, and we are pleased they were satisfied with our revisions. Given that Reviewer 1 and Reviewer 3 both recommended publication in Nature Communications without further revisions, here we only address the final comments of Reviewer 2. Reviewer 2's comments are shown in blue, and changes made to the manuscript noted in orange.

Yours truly, on behalf of all authors,

Nadia Antoniadis, Mark Hogg and Richard Warburton

Reviewer 2

The authors addressed well all the comments. A couple of final points:

1. I agree that fast spin readout is of value to the community, even though it is demonstrated here on a structure with a very strong cotunneling. But I can't agree when the authors say that short electron spin T1 is totally unrelated to the applicability of the readout method. This statement is a speculation until it is proven experimentally. The proximity of the electron Fermi sea, manifested in short T1, has a variety of other possible consequences. One is already appreciated by the authors – the nuclear spin dragging induced by the resonant optical pumping. A thicker barrier, required to achieve a usefully long electron spin qubit T1, may also result in less controllable dynamics of the trapped charges. One may speculate that these and other manifestations of the solid-state environments will not affect the cavity-enhanced readout. In my view this would be a plausible speculation, but speculating is not the same as demonstrating.

I strongly suggest that the authors make this distinction explicit and clear, by explaining that the performance of this readout method in structures with weak cotunneling is subject to further experimental verification. This could be done for example in the paragraph around line 354.

We take Reviewer 2's point here that the ultimate proof that our readout strategy works on a device with weaker cotunneling would be direct experimental evidence. However, we also want to make sure that our text accurately reflects the level of uncertainty regarding the applicability of our readout method to devices with weaker cotunneling. The processes that could impact the cavity-enhanced readout for a device with a thicker tunnel barrier are well understood - we thus believe that describing the applicability of cavity-enhanced readout to devices with weaker cotunneling as "speculation" introduces a level of uncertainty that is unwarranted.

We have added an additional statement to the paper, as suggested by Reviewer 2. The new statement addresses Reviewer 2's concerns, whilst also giving a realistic sense of whether our readout method will work on a device with weaker cotunneling. The new sentence reads as follows (line 356 in the revised manuscript):

"Future experiments using devices with a larger tunnel barrier are strictly necessary to verify that high-fidelity spin initialisation can be combined with cavity-enhanced single-shot readout. Based on the detailed understanding of cotunneling [39], chances of success are very high."

2. "Our open microcavity approach can be used to enhance optical spin readout in other systems, such as nitrogen vacancy centres in diamond [36]". I would argue it has been done already, for example in Science 362, 662 (2018). Moreover, optical cavities have already been used to enhance the readout of the quantum dot spins [Phys Rev Appl 9, 054013 (2018)]. The principle of improving spin readout through cavity-enhanced cyclicity has been applied in other solid state systems, for example in Nature Communications 11, 1605 (2020). I suggest the authors do a bit more work in terms of putting their results into context. This way research done by others would get the due credit, and this paper will have a broader impact.

We accept Reviewer 2's point and have re-phrased the sentence quoted by Reviewer 2. Our original intention was specifically to state that *open microcavities* such as that used in our work can be used in other material platforms. The references cited here by Reviewer 2 all focus on photonic crystal cavities; in comparison, the open microcavity approach is still relatively unexplored. We have rephrased the original sentence, which now reads as follows (line 77 in the revised manuscript):

“Cavity enhancement is a powerful tool for improving optical single-shot spin readout in other systems; photonic crystal cavities have been successfully used with defect centres in diamond [36] and rare-earth ions [37,38]. Importantly our open microcavity approach is not specific to QD samples, and can be used to enhance optical spin readout in other material platforms [39].”